# The effect of evocalcet on vagus nerve activity of the gastrointestinal tract in miniature pigs

**Shin Tokunaga, Takehisa Kawata** [ID]*

R&D Division, Nephrology Research Laboratories, Nephrology R&D Unit, Kyowa Kirin Co., Ltd., Shizuoka, Japan

* takehisa.kawata.kk@kyowakirin.com

## Abstract

Evocalcet is a novel calcimimetic agent with fewer gastrointestinal (GI) adverse effects compared to cinacalcet. Although it is thought that cinacalcet induces GI side effects through the direct stimulation of the calcium receptor (CaR) expressed in the GI tract, the differences in the direct stimulatory effects of these two drugs on the GI tract have not been reported. In this study, we analyzed the difference in the GI effects of these two calcimimetic agents using miniature pigs by detecting vagus nerve stimulation after oral administration of the agents. Although cinacalcet induced vomiting in miniature pigs, evocalcet never induced emetic symptoms. A significant increase in the vagus nerve action potentials was observed after the administration of cinacalcet. Although the increase of that after the administration of evocalcet was mild and not significant in comparison to that in the vehicle group, it was not significantly different from the vagus nerve action potentials after cinacalcet treatment.

## Introduction

Secondary hyperparathyroidism (SHPT), characterized by elevated serum parathyroid hormone (PTH) levels, is a common mineral bone disorder in patients with chronic kidney disease (CKD) [1]. High serum PTH levels often cause abnormal mineral metabolism that is associated with risks of vascular calcification, fracture, and cardiovascular (CV) mortality [2].

Cinacalcet hydrochloride (cinacalcet), a calcimimetic agent that allosterically activates the calcium receptor (CaR) and suppresses PTH secretion, has been widely used to manage SHPT in dialysis patients [3]. Numerous clinical reports have assessed the efficacy of cinacalcet on the treatment of SHPT [4–7]. In these reports, cinacalcet contributed to the adequate control of serum PTH and Ca levels and to reductions in the number of parathyroidectomies (PTx), CV events, fractures, hospitalizations, and mortality in patients with SHPT. However, cinacalcet has adverse effects in the gastrointestinal (GI) tract [8], including nausea and vomiting. It was reported that cinacalcet may be associated with nausea in 32% of patients and with vomiting in 30% of patients in a total of 371 patients [9]. As a result, GI intolerability often limits the dose of cinacalcet and causes poor compliance or discontinuation.

Evocalcet is a novel oral calcimimetic agent that has similar activity to cinacalcet and was developed to reduce the issues associated with cinacalcet use [10–12]. Evocalcet has a higher

**Data Availability Statement:** All relevant data are within the manuscript and its Supporting Information files.

**Funding:** Mitsubishi Tanabe Pharma Corporation provided evocalcet. All studies were performed and the cost of them were supported by Kyowa Kirin

Co., Ltd. The conduct of studies was supported by Masami Kato (Nihon Bioresearch Inc.). Shin Tokunaga and Takehisa Kawata are employees of Kyowa Kirin Co., Ltd. Kyowa Kirin Co., Ltd provided support in the form of salaries for authors, ST and TK, but did not have any additional role in the study design, data collection and analysis, decision to publish, or preparation of the manuscript. The specific roles of these authors are articulated in the 'author contributions' section.

**Competing interests:** Evocalcet is the product in development, and Kyowa Kirin Co., Ltd and Mitsubishi Tanabe Pharma Corporation have the ownership of the patents of evocalcet. These does not alter our adherence to PLOS ONE policies on sharing data and materials.

bioavailability, which contributes to a reduction in the dose compared to that of cinacalcet [13]. Although cinacalcet induced a significant delay in gastric emptying in rats, evocalcet had no such effect [11]. In addition, evocalcet also induced emesis less frequently than cinacalcet in common marmosets [11]. In a clinical trial, the incidence of GI-related adverse effects was lower in the evocalcet group than in the cinacalcet group [14]. Therefore, evocalcet provides a novel therapeutic option for the management of SHPT with fewer GI-related adverse effects.

The mechanism underlying how calcimimetic agents induce GI side effects is still unclear. It was suggested that the activation of the vagus nerve is necessary for inducing GI side effects and vomiting. In this study, we used miniature pigs to observe the stimulatory effect of both drugs on the vagus nerve in order to determine the mechanism underlying the induced GI side effects and compared the stimulatory effects of both drugs. Pigs are similar to humans in eating habits, so the structure of the digestive tract is expected to be physiologically similar to humans [15, 16]. Since the anticancer drug cisplatin also induces vomiting in pigs [17], it is considered a suitable animal for emesis and anti-emetic studies.

## Materials and methods

### Test articles

Cinacalcet was synthesized at Kyowa Kirin Co., Ltd. (Lot No. 504002, Tokyo, Japan). Evocalcet was synthesized at Mitsubishi Tanabe Pharma Corporation (Lot No. 134016, Osaka, Japan). Cisplatin was purchased from Sigma-Aldrich (Lot No. MKBZ9173V, Tokyo, Japan). Evocalcet and cinacalcet were dissolved in 0.5% (w/v) methylcellulose solution, and cisplatin was dissolved in saline.

### Animals

All animal studies were carried out in accordance with the Standards for Proper Conduct of Animal Experiments at Kyowa Kirin Co., Ltd. The protocol was approved by the Institutional Animal Care and Use Committee (IACUC) of the Kyowa Kirin Co., Ltd. (protocol number APS 17J0230-02C and 19J0228). Male NIBS miniature pigs (6–14 months of age) were purchased from Nippon Institute for Biological Science (Tokyo, Japan). Miniature pigs were housed in a room maintained at 20–28˚C with illumination for 12 h. The miniature pigs were allowed free access to water and a NS diet (Nisseiken Co., Ltd., Tokyo, Japan) of approximately 400 g daily. The pigs were acclimated to the environment for at least 1 week prior to the experiment. All animals were monitored once daily throughout the period of the study. The animals in the pharmacodynamics and emesis study were removed from the studies after the completion. In studies of vagus nerve activity, the pigs were fasted overnight before the experiments. The animals in the studies of vagus nerve activity were euthanized by overanesthesia with 5% sodium pentobarbital after the completion of the measurement of vagus nerve potentials.

### Pharmacodynamics study

After acclimatization, the miniature pigs were anesthetized by an intramuscular injection of a combination of medetomidine hydrochloride (0.05 mg/kg) and midazolam (0.5 mg/kg). Anesthesia was subsequently maintained by 1.0%–1.5% isoflurane delivered in gas ($N_2O:O_2 = 1:1$). The catheter was inserted into the sinus venarum cavarum under anesthesia for blood collection. The pigs were divided into three groups matched in terms of their body weight (Group 1: cisplatin, Group 2: cinacalcet, Group 3: evocalcet, n = 2–4/group). Cisplatin was intravenously administered, and evocalcet or cinacalcet were orally administered to each drug treatment group. The blood samples were collected from the catheter before the administration of the

test articles and 30, 120, and 240 minutes after the administration. The serum whole PTH (wPTH) level was measured by radioimmunoassay. The serum calcium levels were measured using an autoanalyzer.

## The evaluation of emesis

Within addition to the pharmacodynamics study, the effects of cisplatin, cinacalcet, and evocalcet on emesis in miniature pigs were simultaneously observed. After the administration of the test agents, the observation of vomiting-like behavior (tongue licking, retching and vomiting) was conducted, and the number of instances of these behaviors was recorded. The observation was carried out for the first 6 h and at the 12, 24, 30, 48, 54, and 72 h time points after the administration.

## The measurement of vagus nerve action potentials

Miniature pigs were anesthetized by an intramuscular injection of ketamine hydrochloride (12 mg/kg) and intravenous injection of thiopental sodium (15 mg/kg). Orotracheal intubation was then performed using a cannula, and animals were ventilated with a respirator. Under anesthesia, a polyethylene catheter was inserted into the femoral artery for the measurement of blood pressure. After completion of the above treatment, the stomach was exposed by an abdominal midline incision. The anterior gastric branches of the anterior vagal nerve trunk were peeled off, and the vagus nerve was cut at the mid-position. Nerve cuff electrodes were placed on each cleavage side of the vagus nerve, and the afferent and efferent potentials were monitored. The bipolar electrode was connected to an amplifier, and the signals were detected and amplified. The obtained signals were digitized as the number of integral spikes (spikes/sec) using the data acquisition system and analyzed using the analysis software program Chart 7 Pro (AD Instruments). Cisplatin was intravenously administered, and evocalcet or cinacalcet were intragastrically administered to each drug treatment group after the stabilization of spikes in vagus nerve. During the experiment, the rectal temperature and blood pressure were monitored. The schematic diagram of the experiment of vagus nerve action potentials is shown in Fig 1.

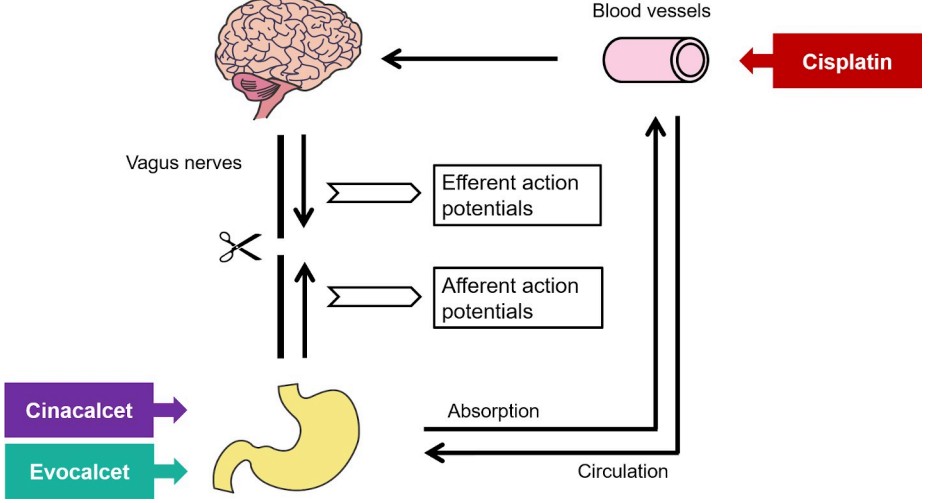

**Fig 1. Schematic diagram of the experiment of the vagus nerve action potentials.**

**Table 1. Serum PTH concentration.**

| Group | Dose (mg/kg) | wPTH (pg/mL) | | |
|---|---|---|---|---|
| | | Pre | 30 min | P value |
| Cinacalcet | 10 | 33.3±8.8 | 21.8±7.1 | 0.365 |
| | 30 | 20.4±5.0 | 11.0±1.1 | 0.129 |
| | 100 | 12.9±0.3 | 11.9±1.8 | 0.605 |
| Evocalcet | 0.3 | 47.6±21.3 | 4.9±0.9 | 0.138 |
| | 1 | 27.9±5.8 | 10.4±1.1 | 0.044 |
| | 3 | 33.4±9.8 | 8.7±0.3 | 0.082 |
| | 10 | 27.3±9.8 | 6.4±0.4 | 0.122 |

Mean±S.E. n = 4/group.

## Statistical analyses

The statistical analyses were all performed using the SAS software program Release 9.2 (SAS Institute Inc., Cary, NC, USA). In the pharmacodynamics study, the differences in the mean values of two points (before and after administration) were determined by a paired $t$-test. In the emesis study, the differences in the mean values (number of instances of vomiting) between the cinacalcet and evocalcet groups were determined by the $\chi^2$ test. In the vagus nerve study, the differences in the mean values among multiple groups were determined by Bartlett's test followed by Tukey's test. P values of <0.05 were considered to indicate statistical significance in all of the analyses.

## Results

### Pharmacodynamic effects

To evaluate the effects on serum PTH and calcium levels, miniature pigs were orally treated with cinacalcet or evocalcet. Cinacalcet tended to reduce the serum PTH level at doses of ≥10 mg/kg and the serum calcium level at doses of ≥30 mg/kg after administration. Evocalcet also tended to reduce the serum PTH levels at doses of ≥0.3 mg/kg and calcium levels at doses of ≥0.3 mg/kg after administration (Tables 1 and 2). However, due to the large variability of individual values, dose dependency could not be confirmed.

### Effects on emesis

To evaluate the emetic effects, 2 miniature pigs were intravenously treated with cisplatin (2 and 5 mg/kg), and 4 were orally treated with cinacalcet (10, 30 and 100 mg/kg) or evocalcet

**Table 2. Serum calcium concentration.**

| Group | Dose (mg/kg) | Calcium (mg/dL) | | | | |
|---|---|---|---|---|---|---|
| | | Pre | 120 min | P value | 240 min | P value |
| Cinacalcet | 10 | 9.9±0.2 | 9.9±0.2 | 0.893 | 9.8±0.1 | 0.297 |
| | 30 | 10.5±0.1 | 10.0±0.1 | 0.076 | 9.7±0.2 | 0.041 |
| | 100 | 10.4±0.1 | 9.5±0.1 | 0.008 | 9.3±0.2 | 0.016 |
| Evocalcet | 0.3 | 9.7±0.2 | 9.4±0.2 | 0.293 | 9.8±0.1 | 0.482 |
| | 1 | 9.4±0.3 | 9.1±0.4 | 0.732 | 9.0±0.5 | 0.668 |
| | 3 | 8.4±0.1 | 8.2±0.1 | 0.287 | 8.0±0.2 | 0.273 |
| | 10 | 10.0±0.2 | 9.2±0.2 | 0.033 | 8.9±0.3 | 0.036 |

Mean±S.E. n = 4/group.

**Table 3. Frequency of vomiting.**

| Group | Dose (mg/kg) | Number of animals | Number of animals with vomiting | P value |
|---|---|---|---|---|
| Cisplatin | 2 | 2 | 2 | NA |
| | 5 | 2 | 2 | NA |
| Cinacalcet | 10 | 4 | 1 | 0.285 |
| | 30 | 4 | 2 | 0.103 |
| | 100 | 4 | 3 | 0.029 |
| Evocalcet | 0.3 | 4 | 0 | Ref |
| | 1 | 4 | 0 | NA |
| | 3 | 4 | 0 | NA |
| | 10 | 4 | 0 | NA |

NA, Not applicable; Ref, reference.

(0.3, 1, 3, and 10 mg/kg). Cisplatin induced vomiting at both doses in 2/2 pigs. Cinacalcet also induced vomiting in 1/4 pigs at 10 mg/kg, 2/4 pigs at 30 mg/kg, and 3/4 pigs at 100 mg/kg. In contrast, no emetic-like symptoms were observed at any dose of evocalcet (Table 3).

### Effects on vagus nerve action potential

To evaluate the effects on the vagus nerve action potential in miniature pigs, animals were treated with cisplatin (2 mg/kg, i.v.), cinacalcet (30 mg/kg, p.o.) or evocalcet (3 mg/kg, p.o.). The total efferent vagus nerve action potentials (%) were 1181.6±324.4 in the vehicle group, 1751.5±163.2 in the evocalcet group, 1845.7±476.7 in the cinacalcet group, and 5733.0±1438.9 in the cisplatin group (Figs 2 and 3). Compared to the vehicle group, a significant increase in total action potential was observed in the cisplatin group; however, no marked increase was observed in the evocalcet or cinacalcet group. In contrast, the total afferent vagus nerve action potentials (%) were 1766.0±524.2 in the vehicle group, 3301.7±260.7 in the evocalcet group, 5582.1±1189.7 in the cinacalcet group, and 5838.3±1537.1 in the cisplatin group (Figs 4 and 5). Compared with the vehicle group, significant increases in total afferent action potentials were observed in the cinacalcet and cisplatin groups. Although the total action potential in the evocalcet group was increased compared to that in the vehicle group, it was not significant

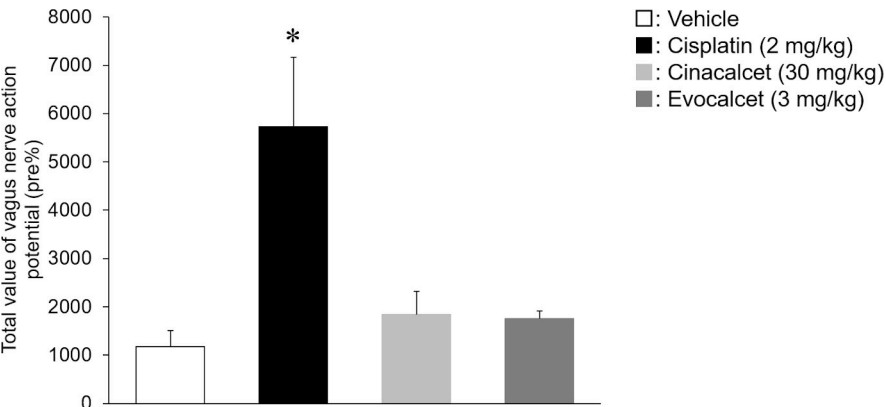

**Fig 2. Total value of the efferent vagus nerve action potentials rate of change between the groups.** The total of efferent vagus nerve action potentials for 2 hours after administration were calculated. The data are presented as the mean + S.E. n = 6–8/group. * P < 0.05 vs. Vehicle group (Tukey's test).

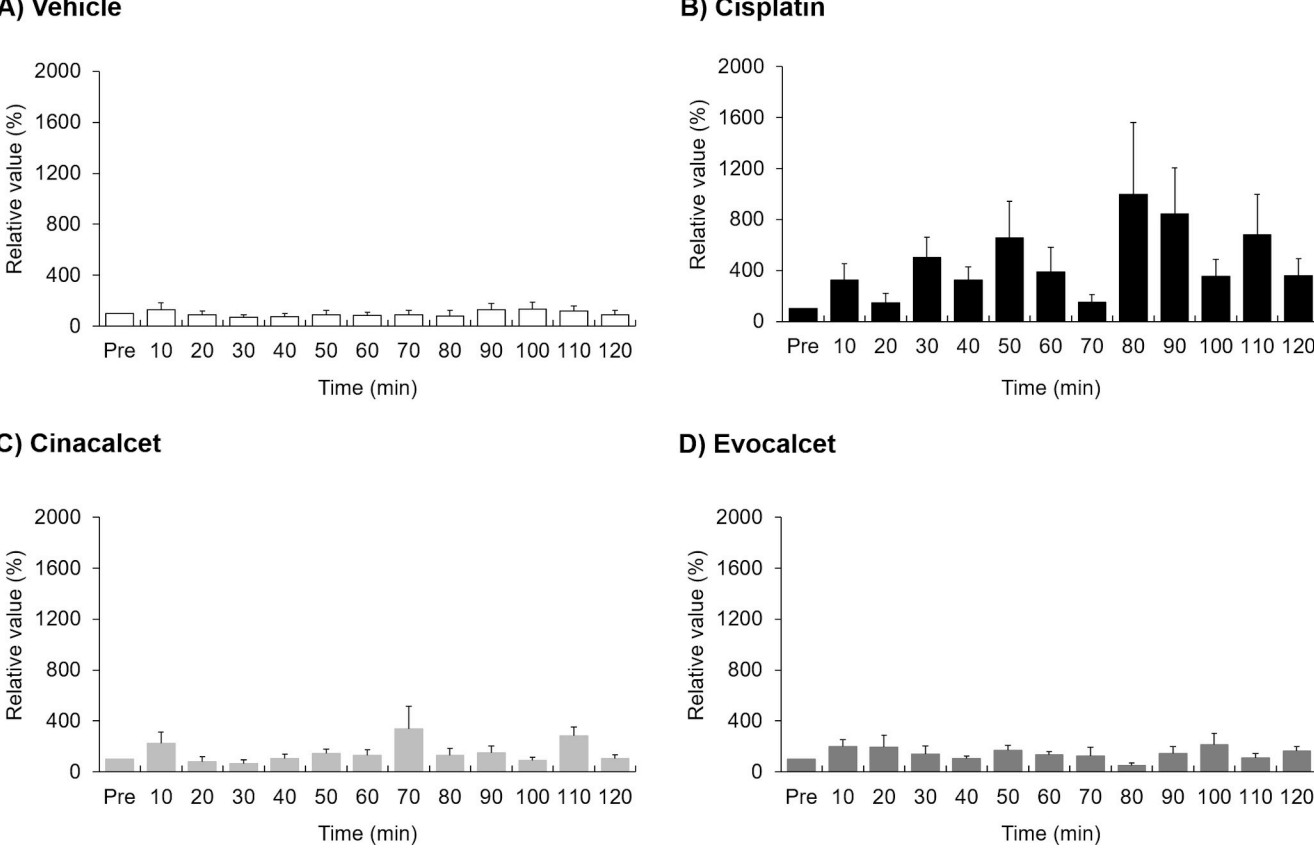

**Fig 3. The efferent vagus nerve action potential rate of change at each time points.** The data are presented as the mean + S.E. n = 6–8/group. (A) Vehicle, (B) Cisplatin, (C) Cinacalcet, (D) Evocalcet.

(p = 0.536) and was lower than in the cinacalcet group. However, the total afferent action potentials of the cinacalcet and evocalcet groups did not differ to a statistically significant extent (p = 0.587). These increases in afferent vagus nerve action potentials were observed

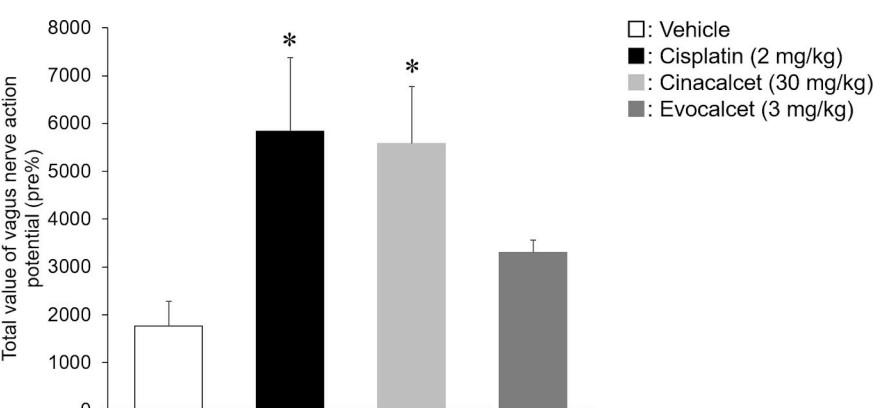

**Fig 4. Total value of the afferent vagus nerve action potentials rate of change between the groups.** The total of afferent vagus nerve action potentials for 2 hours after administration were calculated. The data are presented as the mean + S.E. n = 6–8/group. * P < 0.05 vs. Vehicle group (Tukey's test).

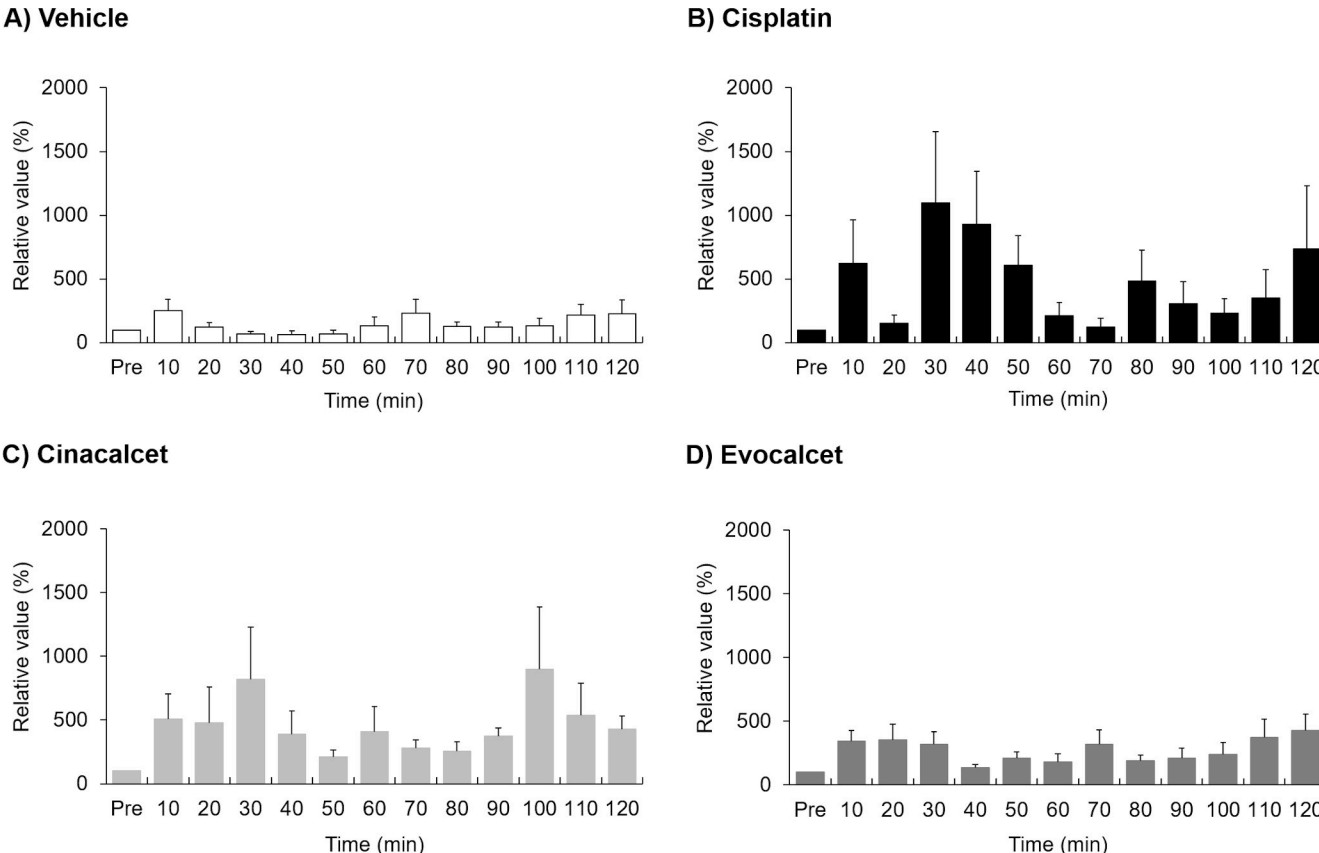

**Fig 5. The afferent vagus nerve action potential rate of change at each time points.** The data are presented as the mean + S.E. n = 6–8/group. (A) Vehicle, (B) Cisplatin, (C) Cinacalcet, (D) Evocalcet.

from the early phase 10 minutes after administration and continued until the end of the observation periods at 120 minutes after administration (Fig 5).

## Discussion

Although cinacalcet effectively suppresses the PTH and calcium levels in patients with SHPT, it frequently induces GI events, which often results in poor adherence. Evocalcet, a recently developed new oral calcimimetic agent, was designed to ameliorate these GI events [11], and its efficacy was shown in clinical trials [14].

In the present study, to estimate the direct effect of both drugs on the GI tract, we compared the stimulatory effects of both drugs on the vagus nerve action potentials in miniature pigs. The GI tract of a miniature pig is considered similar to that of a human, both anatomically and physiologically [15, 16]. Since cisplatin induces acute and delayed vomiting in miniature pigs, miniature pigs seem to be an appropriate model for the evaluation of GI events. In our study, cisplatin was used as an emetic positive control drug because acute and delayed vomiting have been observed in treatment with cisplatin. It was also reported that the ablation of the vagus nerve significantly suppressed vomiting induced by cisplatin [18], so the activation of the vagus nerve seems to be necessary for inducing vomiting. In this study, the administration of cisplatin induced an increase in the total action potential at both the afferent and efferent vagus nerves. Therefore, it was suggested that the increase in the vagus nerve action potentials

may be associated with the mechanism of vomiting, and cisplatin-induced vomiting by direct stimulation to both the central nerves and the GI tract.

The oral administration of cinacalcet increased the afferent vagus nerve action potential toward the central nerves from the GI tract. This increase in afferent action potentials occurred early after the administration and continued until the end of the observation period (120 minutes after administration). We suspect that cinacalcet activated the CaR expressed in the GI tract without an elevation in its serum concentration, as this increase occurred early after treatment. Since there is a report of cinacalcet inhibiting gastric emptying in rats [11], it was also suggested that cinacalcet inhibited gastric emptying in miniature pigs, allowing the drug to remain in the stomach for a long time, which contributed to the continuous stimulation of CaR expressed thereon and the elevation of afferent action potentials. In contrast, no clear change in the efferent vagus nerves action potentials with respect to the central nerves to the upper part of the GI tract was noted. This result suggested that cinacalcet never directly stimulates the central nerves. Although it is difficult to elucidate the mechanism of nausea and vomiting induced by calcimimetics, GI symptoms associated with hypercalcemia has been reported in various reports [19–21], which is an important knowledge for understanding. For example, it was reported the GI symptoms (anorexia, nausea, vomiting, abdominal pain, and constipation) occurred in 66% of primary hyperparathyroidism patients with hypercalcemia [22]. Details concerning how cinacalcet stimulates the GI tract remain unclear; however, it is likely that the GI side effects induced by cinacalcet are caused by the direct stimulation of CaR expressed in the GI tract.

In the pharmacodynamics study, cinacalcet reduced the serum PTH levels at doses of $\geq 10$ mg/kg, and evocalcet also decreased the serum PTH levels at doses of $\geq 0.3$ mg/kg. In this dose range, cinacalcet induced acute or delayed vomiting at doses of 10, 30, and 100 mg/kg, but the animals who received evocalcet did not show any symptoms of vomiting at a dose of 10 mg/kg. Cinacalcet induced vomiting from the pharmacological effective dose, whereas evocalcet did not induce vomiting, even at 30 times the effective dose. In the vagus nerve study, although evocalcet showed a milder effect on the afferent vagus nerve action potentials, the afferent vagus nerve action potentials in the cinacalcet and evocalcet groups did not differ to a statistically significant extent. We considered some possible reasons as to why our hypothesis, that evocalcet would have a weaker effect on the afferent vagus nerve action potentials in comparison to cinacalcet, was not proven in this study. As the first reason, in the present study, the numbers of animals and/or experiments might have been too small; thus, the study may have lacked the power to demonstrate statistical significance. Additionally, the vagus nerve study was performed under anesthesia, while the emesis study and clinical trials were performed under arousal. Anesthesia is thought to affect the duration for which the drug remains in the GI tract. In the vagus nerve study, the drug would be more likely to remain in the GI tract longer under anesthesia, and the evocalcet was more likely to have a stimulatory effect in comparison to experiments performed under arousal.

The differences in the GI side effects between evocalcet and cinacalcet may be due to the differences in the dose of each drug. It was reported that the bioavailability of evocalcet in rats was more than 80%, whereas that of cinacalcet is approximately 1%–2% [11]. The reduction in the pharmacologically effective dose of evocalcet due to its higher bioavailability appears to contribute to the reduced direct stimulation of the GI tract, since of the dose administered, only a small amount of evocalcet is exposed in the stomach.

Additionally, it has been reported that evocalcet has less of an effect on gastric emptying compared to cinacalcet. The rapid disappearance of evocalcet from the digestive tract with its good bioavailability also might lead to limited stimulation of the efferent vagal nerves. On the other hand, cinacalcet is known to delay gastric emptying, resulting in a prolonged stay in the

GI tract and sustained efferent vagal stimulation. Evocalcet is also reported to have a wider safety margin for emesis than cinacalcet in common marmosets [11]. In the clinical study, the GI side effects of evocalcet were also decreased compared to those of cinacalcet [14]. These results support the notion that evocalcet stimulates the GI tract to a lower degree than cinacalcet.

## Conclusions

The present study confirmed that cinacalcet caused GI side effects through the direct stimulation of the GI tract, which was associated with an elevation of the afferent vagus nerve action potentials. Evocalcet—a novel calcimimetic agent with an improved pharmacokinetic profile in comparison to cinacalcet—did not induce vomiting in miniature pigs, and tended to show a mild effect on the afferent vagus nerve action potentials in comparison to cinacalcet, although these differences were not statistically significant. Our results therefore suggest that evocalcet may be a potent alternative to existing calcimimetics for the management of SHPT, with a lower rate of GI-related adverse events.

## Supporting information

**S1 Table. The set of raw data for Table 1.**
(XLSX)

**S2 Table. The set of raw data for Table 2.**
(XLSX)

**S3 Table. The set of raw data for Fig 2.**
(XLSX)

**S4 Table. The set of raw data for Fig 3.**
(XLSX)

**S5 Table. The set of raw data for Fig 4.**
(XLSX)

**S6 Table. The set of raw data for Fig 5.**
(XLSX)

## Author Contributions

**Conceptualization:** Shin Tokunaga, Takehisa Kawata.

**Data curation:** Shin Tokunaga.

**Investigation:** Takehisa Kawata.

**Methodology:** Shin Tokunaga, Takehisa Kawata.

**Visualization:** Shin Tokunaga.

**Writing – original draft:** Shin Tokunaga.

**Writing – review & editing:** Takehisa Kawata.

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
