## [Decision Letter · Decision Letter 0]

11 Nov 2020

PONE-D-20-16625

The effect of evocalcet on vagus nerve stimulation of the gastrointestinal tract in miniature pigs

PLOS ONE

Dear Dr. kawata,

Thank you for submitting your manuscript to PLOS ONE. After careful consideration, we feel that it has merit but does not fully meet PLOS ONE’s publication criteria as it currently stands. Therefore, we invite you to submit a revised version of the manuscript that addresses the points raised during the review process.

We look forward to receiving your revised manuscript.

Kind regards,

Julia Robinson

Senior Editor

PLOS ONE

Journal Requirements:

2. Thank you for providing the following Funding Statement: 

'Mitsubishi Tanabe Pharma Corporation provided evocalcet. All studies were performed and the cost of them were supported by Kyowa Kirin Co., Ltd. Shin Tokunaga and Takehisa Kawata are employees of Kyowa Kirin Co., Ltd. Kyowa Kirin Co., Ltd provided support in the form of salaries for authors, ST and TK, but did not have any additional role in the study design, data collection and analysis, decision to publish, or preparation of the manuscript.'

We note that one or more of the authors is affiliated with the funding organization Kyowa Kirin Co., Ltd., indicating the funder may have had some role in the design, data collection, analysis or preparation of your manuscript for publication; in other words, the funder played an indirect role through the participation of the co-authors.

a. If the funding organization did not play a role in the study design, data collection and analysis, decision to publish, or preparation of the manuscript and only provided financial support in the form of authors' salaries and/or research materials, please review your statements relating to the author contributions, and ensure you have specifically and accurately indicated the role(s) that these authors had in your study in the Author Contributions section of the online submission form. Please make any necessary amendments directly within this section of the online submission form. 

Please also update your Funding Statement to include the following statement: “The funder provided support in the form of salaries for authors [insert relevant initials], but did not have any additional role in the study design, data collection and analysis, decision to publish, or preparation of the manuscript. The specific roles of these authors are articulated in the ‘author contributions’ section.”

If the funding organization did have an additional role, please state and explain that role within your Funding Statement.

We also note that you have a patent relating to material pertinent to this article.

Please declare this patent (with details including name and number), along with any other relevant declarations relating to employment, consultancy, patents, products in development or modified products etc. in the amended statement of Competing Interests.

Please confirm that this does not alter your adherence to all PLOS ONE policies on sharing data and materials, as detailed online in our guide for authors http://journals.plos.org/plosone/s/competing-interests by including the following statement: "This does not alter our adherence to  PLOS ONE policies on sharing data and materials.” If there are restrictions on sharing of data and/or materials, please state these.

Please note that we cannot proceed with consideration of your article until this information has been declared.

Reviewers' comments:

Reviewer's Responses to Questions

**Comments to the Author**

1. Is the manuscript technically sound, and do the data support the conclusions?

Reviewer #1: Yes

Reviewer #2: Yes

Reviewer #3: Yes

2. Has the statistical analysis been performed appropriately and rigorously? 

Reviewer #1: Yes

Reviewer #2: Yes

Reviewer #3: Yes

3. Have the authors made all data underlying the findings in their manuscript fully available?

Reviewer #1: Yes

Reviewer #2: Yes

Reviewer #3: Yes

4. Is the manuscript presented in an intelligible fashion and written in standard English?

Reviewer #1: Yes

Reviewer #2: Yes

Reviewer #3: Yes

5. Review Comments to the Author

Reviewer #1: Manuscript Number: PONE-D-20-16625

The effect of evocalcet on vagus nerve stimulation of the gastrointestinal tract in　miniature pigs

The authors analyzed the incidences of emetic symptoms induced by the administrations of cinacalcet or evocalcet in miniature pigs. Cinacalcet induced the symptoms dose-dependently, however, evocalcet did no symptom. They also examined the vagus nerve stimulation of the gastrointestinal tract by these two calcimimetics, and found that action potential of afferent vagus induced the symptom.

The experiments are well-planned and the results themselves are well-understood. Cisplatin, employed as active placebo to compare the effects of emetic symptoms and vagus nerve stimulation to calcimimetics, made their results very clear.

Although the authors concluded the emetic symptoms were evoked by CaR stimulation in stomach, the evidences were still weak, which they mentioned in the discussion. Actually hypercalcemia caused by primary hyperparathyroidism evokes many symptoms including gastrointestinal symptoms such as nausea/vomiting. These clinical evidences should be disucussed.

Page 3, line 36 “critical” is too strong, milder expressions would be suitable for this sentence.

Page 3, line 43 “higher bioavailability” needs references

Page 3, line 45 “no such effect” needs references.

Page 3, line 46 “in common marmosets” needs references.

Page 4, line 52 Why used miniature pigs? Kindly give an explanation and references.

Page 4, line 65 The expression “strict accordance” is wired. delete “strict.”

Page 5, line 77 The expression “overanesthesia” is wired. “Overanesthesia” may kill an animal.

Table 1, table 2, table 3 Kindly give P values in these tables.

Figure 3, figure 5 The time-course changes in these compound were significant?

Figure legends of fig 5

“The efferent vagus nerve” should be “The afferent vagus nerve.”

Reviewer #2: An interesting study and my comments are minor.

1. Title: Suggest changing to "vagal nerve activity" as you are studying the effects of the drugs on vagal nerve activity "at rest" not during stimulation

2. Method: Your observations of vomiting-like behaviours includes vomiting. Better to say that you quantified the numbers of vomits (ie. oral expulsion of gastric contents), dry retching and additional behaviours associated with vomiting.

3. When compounds were administered intragastrically was this via the esophagus or an incision in the stomach?

4. What were the statistical significance values for the initial serum PTH/calcium measurements?

5. Elsewhere, when you talk about differences which tended to occur but which were not statistically significant, give the P value.

6. Since there are other ways of causing vomiting, apart from vagal nerve activation (eg. by stimuli in the blood activating neurons in the area postrema), it would be good to say that you focussed on the vagus and not this other route because of the low oral bioavailability of cinacalcet. In addition, since the authors drug has good oral bioavailability (and therefore likely to access the area postrema), the absence of vomiting could be used to add further support to the authors argument that it is the retention of cinacalcet within the upper GI tract that leads to the GI side effects

Reviewer #3: Authors clarified the mechanism of GI tract adverse effects of cinacalcet to measure the action potential of the vagus nerve after oral administration in miniature pigs and found the significant difference of the afferent nerve between cinacalcet and Evocalcet, a novel calcimimetic agent..

1) Based on these findings, does high concentration of calcium ion increase action potential of the afferent vagus nerve and induce vomiting through the CaR stimulation in the GI tract? Please discuss.

2) In Fig 2 and Fig 4, why did authors choose 2hr treatment of the agents, instead of 0.5-1hr treatment?

3) In Fig 5, did authors compare delta value of action potentials between (Evo-Veh) and (Cin-Veh) in acute phase (10-60 min)? If you find statistically significant difference, you may add new figure and stress the difference of these two agents.

4) In Line 45 and Line 46, the references are necessary.

6. PLOS authors have the option to publish the peer review history of their article (what does this mean?). If published, this will include your full peer review and any attached files.

Reviewer #1: No

Reviewer #2: No

Reviewer #3: No

---

## [Author Response · Author response to Decision Letter 0]

22 Dec 2020

We thank very much for the carefully reviewing our manuscript and the fruitful suggestions, especially for suggesting better terms and sentences concerning our manuscript. We found the comments of the reviewers most helpful and thus revised our manuscript according to the suggestions of the reviewers. We enclose revised versions of the manuscript, both with highlights of all changes and with unmarked changes. We also include a letter of our itemized responses to the Editor’s and Reviewer’s comments. We hope that this revised manuscript is now acceptable for publication in PLOS ONE.

Thank you again for considering our manuscript. We are looking forward to hearing from you soon.

---

## [Decision Letter · Decision Letter 1]

8 Jan 2021

The effect of evocalcet on vagus nerve activity of the gastrointestinal tract in miniature pigs

PONE-D-20-16625R1

Dear Dr. takehisa kawata

We’re pleased to inform you that your manuscript has been judged scientifically suitable for publication and will be formally accepted for publication once it meets all outstanding technical requirements.

Kind regards,

Olivier Barbier

Academic Editor

PLOS ONE

Reviewers' comments:

Reviewer's Responses to Questions

**Comments to the Author**

1. If the authors have adequately addressed your comments raised in a previous round of review and you feel that this manuscript is now acceptable for publication, you may indicate that here to bypass the “Comments to the Author” section, enter your conflict of interest statement in the “Confidential to Editor” section, and submit your "Accept" recommendation.

Reviewer #1: All comments have been addressed

Reviewer #2: All comments have been addressed

Reviewer #3: All comments have been addressed

2. Is the manuscript technically sound, and do the data support the conclusions?

Reviewer #1: Yes

Reviewer #2: (No Response)

Reviewer #3: Yes

3. Has the statistical analysis been performed appropriately and rigorously? 

Reviewer #1: Yes

Reviewer #2: (No Response)

Reviewer #3: Yes

4. Have the authors made all data underlying the findings in their manuscript fully available?

Reviewer #1: Yes

Reviewer #2: (No Response)

Reviewer #3: Yes

5. Is the manuscript presented in an intelligible fashion and written in standard English?

Reviewer #1: Yes

Reviewer #2: (No Response)

Reviewer #3: Yes

6. Review Comments to the Author

Reviewer #1: (No Response)

Reviewer #2: (No Response)

Reviewer #3: Authors made suitable correction of the manuscript. Although authors did not address this issue, in the future, please clarify if high concentration of calcium ion increase action potential of the afferent vagus nerve and induce vomiting through the CaR stimulation in the GI tract using this model.

7. PLOS authors have the option to publish the peer review history of their article (what does this mean?). If published, this will include your full peer review and any attached files.

Reviewer #1: No

Reviewer #2: No

Reviewer #3: **Yes: **Shozo Yano

---

## [Editor Report · Acceptance letter]

12 Jan 2021

PONE-D-20-16625R1 

The effect of evocalcet on vagus nerve activity of the gastrointestinal tract in miniature pigs 

Dear Dr. Kawata:

I'm pleased to inform you that your manuscript has been deemed suitable for publication in PLOS ONE. Congratulations! Your manuscript is now with our production department. 

Kind regards, 

on behalf of

Prof. Olivier Barbier 

Academic Editor

PLOS ONE